# Mapping Research Trends from 20 Years of Publications in Rhythmic Auditory Stimulation

**DOI:** 10.3390/ijerph20010215

**Published:** 2022-12-23

**Authors:** Meiqi Zhang, Fang Li, Dongyu Wang, Xiaohong Ba, Zhan Liu

**Affiliations:** 1Department of Physical Education and Health Education, Springfield College, Springfield, MA 01109, USA; 2Yale/VA Learning-Based Recovery Center, Yale University, New Haven, CT 06510, USA; 3Department of Neurology, The First Affiliated Hospital of Jinzhou Medical University, Jinzhou 121001, China; 4Department of Neurology, The Center Hospital of Jinzhou, Jinzhou 121001, China

**Keywords:** auditory cueing, motor dysfunction, music therapy, Parkinson’s disease

## Abstract

This study aims to create an all-around insight into the evolutions, status, and global trends of rhythmic auditory stimulation (RAS) research via enhanced bibliometric methods for the 2001–2020 time period. Articles concerning RAS were extracted from the Web of Science database. CiteSpace, Bibliometrix, VOSviewer, and Graphpad Prism were employed to analyze publication patterns and research trends. A total of 586 publications related to RAS between 2001 and 2020 were retrieved from the Web of Science database. The researcher Goswami U. made the greatest contribution to this field. The University of Toronto was the institution that published the most articles. Motor dysfunction, sensory perception, and cognition are the three major domains of RAS research. Neural tracking, working memory, and neural basis may be the latest research frontiers. This study reveals the publication patterns and topic trends of RAS based on the records published between 2001 and 2020. The insights obtained provided useful references for the future research and applications of RAS.

## 1. Introduction

Rhythmic auditory stimulation (RAS) is a therapeutic technique that uses predictable time cues to improve different outcomes related to movement [1,2]. RAS is characterized by acoustic stimuli that synchronize motor response through rhythmic cueing, and the synchronous connection is produced by neuronal substrates with effects on movements, therefore the synchronization is created between the subject’s movements and external auditory stimuli through an internal timing process [3,4,5]. Consequently, the subject will perform the time-coordinated movements and the rhythmic sequence programmed by the RAS model [4,6]. 

A large number of clinical research and systematic reviews have examined the effectiveness of RAS in the rehabilitation of Parkinson’s disease (PD) [5,7], stroke [8], language disorders [9], and other sensorimotor dysfunctions. Combined strategies (e.g., integrating auditory cues and visual stimuli) have also been investigated to optimize the use of the technique.

Bibliometric analysis is used for analyzing literature characteristics, research impact, and trends over time [10]. In comparison to other reviewing methods, bibliometrics provide a more quantitative way of measuring research impact, so are seen as relatively objective. Bibliometric analysis has been widely applied to public health, rehabilitation, and medical fields [11,12,13]. A growing number of practitioners have focused on conducting RAS research over the past two decades. However, a report on RAS using bibliometric analysis has not been found yet.

This study conducts a bibliometric analysis of RAS based on articles published between 2001 and 2020 to provide an all-around insight into the evolutions, status, and global trends of RAS research. The results can also be used to facilitate further experimental designs, study submissions, and clinical applications.

## 2. Methods

### 2.1. Source of Data and Search Strategy

Published papers were retrieved via a topic search of the Science Citation Index Expanded (SCI-Expanded) from the Web of Science Core Collection database (WoSCC) in September 2021. The searching strategy was based on free-terms: TS = (rhythmic + auditory + cue* OR rhythmic + auditory + stimulation* OR rhythmic + auditory + training* OR rhythmic + auditory + therapy* + rhythmic + acoustic + stimulation*), index = SCI-Expanded, and time span = 2001–2020.

### 2.2. Inclusion Criteria

Only studies designated as “article” were included, covering all types of studies. Other document types such as meeting abstracts and letters, were excluded. The critical information (i.e., titles, abstracts, journals, countries/regions, institutions, and cited references) were comprehensively collected. Only studies that were published in English were considered. Finally, a total of 586 records published during the period 2001–2020 met the inclusion criteria.

### 2.3. Analytical Methods

The data were analyzed by three bibliometric analysis tools: Bibliometrix R Package [14], VOSviewer 1.6.11 [15], Citespace [16], and GraphPad Prism 8 for Windows [13]. Bibliometrix was conducted to analyze the frequency distribution of annual outputs, authors, articles, journals, countries, collaboration between countries, and institutions according to the titles and abstracts. VOSviewer 1.6.11 [15] was used to design the social network maps for co-occurrence keywords that appeared in at least 10 papers. Citespace was applied to detect the keywords with citation bursts. A citation burst has two characteristics: strength and duration [12,17]. A citation burst indicates increased attention to the underlying work over a certain period, which is a key indicator for determining emerging trends [18,19,20]. GraphPad Prism 8 was applied to analyze the time trend of annual publication outputs.

## 3. Results

### 3.1. Literature Distribution

A total of 586 articles were identified. As Figure 1 shows, since 2001, the number of publications concerning RAS has risen considerably. A sharp increase was observed in 2014, which the number of articles surged from 29 in 2013 to 69 in 2014. Then, the annual outputs held steady at around 50 articles.

Research relating to RAS has been published in 234 journals. *PLoS One* published the most RAS-relevant studies, with 32 articles (Table 1). The *Frontiers in Neuroscience* ranked second with 20 articles. The *Experimental Brain Research*, *Frontiers in Human Neuroscience*, *Human Movement Science*, *Journal of Neuroscience,* and *Neuroimaging* tied for the third, publishing 16 articles on RAS research, respectively. The *Frontiers in Neurology* (*n* = 15), *European Journal of Neuroscience* (*n* = 14), and *Gait & Posture* (*n* = 14) ranked from eighth to tenth. Amongst the ten journals, there were six neuroscience journals, one multidisciplinary journal, one psychological journal, one neuroimaging journal, and a rehabilitation journal.

Authors from 40 countries have contributed to research on RAS. As shown in Figure 2, the United States was the most productive country in RAS research and ranked the first in publication outputs, accounting for around 30% (*n* = 176) of the total. The United Kingdom (*n* = 60), Germany (*n* = 54), Canada (*n* = 53), Italy (*n* = 33), France (*n* = 30), The Netherlands (*n* = 25), Australia (*n* = 24), South Korea (*n* = 23), and China (*n* = 22) ranked from second to tenth and were all far behind on article totals. Figure 3 shows the collaborative relationships of the Top 20 most productive countries within RAS research. It is obvious that the United States plays an irreplaceable leading role, which has built relatively stable collaborative relationships with at least ten countries. What should also be noticed is that the UK and Germany also demonstrated the high centrality to collaborative research in RAS. The collaborations displayed a significantly regional characteristic.

Aligned with the contribution of countries, Table 2 shows the 10 most contributive institutions in the RAS research field. The University of Toronto was the most productive institution, which had 42 publications related to RAS. The Colorado State University and Radboud University Nijmegen tied for second with 26 publications. They were followed by the University of Cambridge (*n* = 25), Northwestern University (*n* = 22), Washington University (*n* = 22), McMaster University (*n* = 21), University of Maryland (*n* = 21), University of California Irvine (*n* = 20), and Max Planck Institute for Human Cognitive and Brain Sciences (*n* = 18). Half of the institutions were American colleges. The other five included two Canadian institutions, and The Netherlands, Germany, and the UK each had one.

A total of 2263 authors contributed to the total output of RAS research. The top 10 authors published a total of 101 papers (Table 3), which accounted for 17.23% of all the literature in this field. Dr. Usha Goswami from the University of Cambridge published the most research on RAS with 15 papers, followed by Dr. Michael H. Thaut from the University of Toronto and Dr. ‪Simone Dalla-Bella from the University of Montreal, who had 13 and 12 articles, respectively. Ranking from fourth to tenth were Nieuwboer A. (*n* = 10), Rochester L (*n* = 10), Kotz S. (*n* = 9), Trainor L. (*n* = 9), Grahn J. (*n* = 8), Kwakkel G. (*n* = 8), and Earhart G. (*n* = 7).

### 3.2. Research Topics and Hotspots

The keywords co-occurrence analysis produced a network map to reflect high-frequency keywords and topics within RAS research. As Figure 4 shows, there are 3 clusters for 60 high-frequency keywords within the field of RAS research. The keywords in cluster 1 (colored by red) are mainly related to movement dysfunctions, cluster 2 (colored by blue) covers keywords related to perception, and cluster 3 (colored by gold) tends to be more related to cognition. This represents that previous research is mainly based on these three directions.

The frequency of citations could reflect the research topics within the field. The top ten cited articles in the field of RAS were shown in Table 4. Among the ten articles, there were two stroke studies and two PD studies, and the other six were related to perception and cognitive process. The article that received the highest citation, “Rhythm and beat perception in motor areas of the brain”, was published in the *Journal of Cognitive Neuroscience* in 2007 and received a total of 501 citations. Most of the articles (i.e., 7/10) were published in journals in the field of neuroscience, and the other three appeared in medical and multidisciplinary journals. The top ten cited RAS articles in the past five years (2016–2020) were shown in Table 5. The ten articles tended to focus more on the neural mechanisms of RAS. The most cited article was “Motor origin of temporal predictions in auditory attention”, which was published in *Proceedings of the National Academy of Sciences of the United States of America* in 2017. Among these ten articles, four were published in neuroscience journals, four in multidisciplinary journals, one in a biological journal, and one in a psychological journal.

Burst detection was applied to investigate the research hotspots further. Burst keywords can be identified as indicators of emerging trends [42]. Figure 5 presents keywords with the 25 strongest citation bursts between 2011 and 2020. The most recent burst keywords were working memory, tracking, and neural basis.

Clinically, RAS has been widely applied as a rehabilitation strategy. According to the analysis of high-frequency keywords, PD was the most focused disease within RAS research (Figure 6). A total of 97 publications focused on PD. Stroke and speech impairment ranked second and third with 50 and 47 publications, while the other six were well below: dyslexia (18 articles), dementia (18 articles), multiple sclerosis (10 articles), apraxia (6 articles), cognitive impairments (6), cerebral palsy (5 articles), and depression (5 articles). All the diseases were neurological and psychiatric disorders.

## 4. Discussion

The bibliometric analysis aimed to investigate the developmental trends and hotspots of research on 586 RAS articles during 2001–2020 by using VOSviewer, Bibliometrix, and Citespace. This work summarized research trends, topics, sources, and contributions for RAS.

An exponential growth has been observed in the output of RAS research. If the overall growth rate that was observed also applies to the future, we can expect the publication volume in the RAS research field to double about every five years. This strong growth of research output may be partially attributed to the increased prevalence of neurologic and psychiatric diseases (those were major types of disease applying RAS treatment). Given this, researchers and practitioners worked on exploiting RAS to facilitate rehabilitation. Intervention methods and analysis strategies are being developed [16,43,44]. The surge of RAS research in 2014 might be influenced by the release of the Global Burden of Disease (GBD) 2010 study at the end of 2012 [45]. The GBD 2010 found 114% and 68% increases in the mortality of psychiatric and neurological disorders from 1990 [46,47,48,49].

Among the top-performing journals, more than half were neuroscience journals. It seems that RAS researchers preferred to submit studies to the neuroscience area. An explanation is that for promoting research in the relevant field, some of the neuroscience journals have set auditory cognition and/or brain stimulation research sections or special columns. Most RAS studies were published in multidisciplinary journals also reflected a trend. This could be attributed to how the multidisciplinary journals tended to have a larger number of articles published than subject-specific journals. In addition, RAS has always been considered a transdisciplinary subject [50]. Publishing in multidisciplinary journals may expand the influence of studies.

Of the top productive countries, the U.S. ranked first in the number of RAS publications. Five American affiliations came in the top 10 institutions in the research of RAS, symbolizing that the U.S. possessed the most elite institutions around the world. However, there is only one American author in the top ten authors, indicating the contributions of US researchers were more even. This may be because the concept of integrating RAS into rehabilitation was initially proposed by an American researcher [51] thus the U.S. went deeper in this field as compared with the rest of the world. It is also worth mentioning that the boost of RAS clinical research should be attributed to a Canadian researcher, Dr. Michael Thaut, who has revolutionized the use of auditory stimulation in rehabilitation around the world.

Keyword analyses can be used for detecting research topics, analyzing research hotspots, as well as monitoring the research frontier transitions [52]. Through the analysis of the high-frequency keywords in the RAS research of the past 20 years, the studies were closely related to the three themes: motor dysfunction, cognition, and perception. This hypothesis was also supported by burst detection.

### 4.1. Motor Dysfunctions

Facilitating walking rehabilitation is a major use of RAS [7,32]. RAS demonstrated significant effects on improving muscle coordination and balance performance to patients with PD and stroke [53,54]. Early researchers focused more on the patterns of auditory cues [4,6]. More recently, researchers began to investigate the integrated effects of RAS and other techniques, such as exercise and robot-assisted therapeutic devices [55]. The comprehensive efficacy of RAS is constantly being verified in clinical trials for the rehabilitation of various movement disorders/dysfunctions [7,56,57,58].

### 4.2. Cognition

Early studies regarding the beneficial effects of auditory stimuli on cognition (e.g., learning and memory) had been inconsistent [59]. However, more recent neurobiological research revealed the significant roles of RAS for cognitive performance of Alzheimer’s disease, developmental language disorders, and also PD [9,23,45,60]. To boost cognitive processing, a temporal sampling framework for specific disorders can be integrated into auditory stimuli [9]. Additionally, bass sounds were examined to be better at allocating attention than high-frequency sounds, providing incomplete evidence for the convention of using bass instruments to carry the rhythmic foundations of music [61].

### 4.3. Perception

The broad application of electrophysiological technologies (e.g., electroencephalogram) allowed researchers to explore the associations between RAS and sensory perception. Most early studies attempted to explain how neural sensory, and motor contributed to behavioral beat synchronization [62,63]. Over time, neural tracking, the mechanism of neural signals synchronizing to rhythmic sensory inputs [64], has been found to excel in rhythmic coordination of perception and action [61]. Clinically, RAS has been examined to be effective on improving perceptual performance, especially speech and visual perception [45,46].

We tried to provide information for future research and clinical uses based on quantitative bibliometrics, but a few limitations should be noticed. One is the findings of the study may be limited by the scope of the analytic tools. We only retrieved data from the WoS database, some vital information might have been missed. Another one is that the bibliometric data might not be adequate to assess the actual contribution of the research/researchers. Even though the quantitative metrics (e.g., number of publications) can reflect the popularity, the results should be explained carefully.

## 5. Conclusions

This study may help practitioners and researchers discover the publication patterns and emerging trends of RAS and present reference values for future research and applications. The most influential author, institutions, journals, and countries were Goswami U., University of Toronto, PLoS ONE, and the U.S., respectively. Motor dysfunction, sensory perception, and cognition are the three major clinical domains of RAS research. Neural tracking, working memory, and neural basis may be the latest research frontiers.

## Figures and Tables

**Figure 1 ijerph-20-00215-f001:**
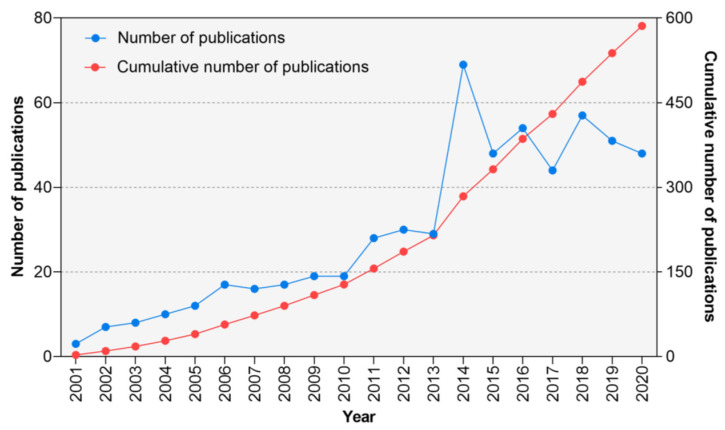
Number of publications and cumulative number of publications on rhythmic auditory stimulation publication. Alt text: A line graph plots the number of annual publications and cumulative number of publications among 2001–2020. A generally exponential growth is observed. A surge is observed in 2014.

**Figure 2 ijerph-20-00215-f002:**
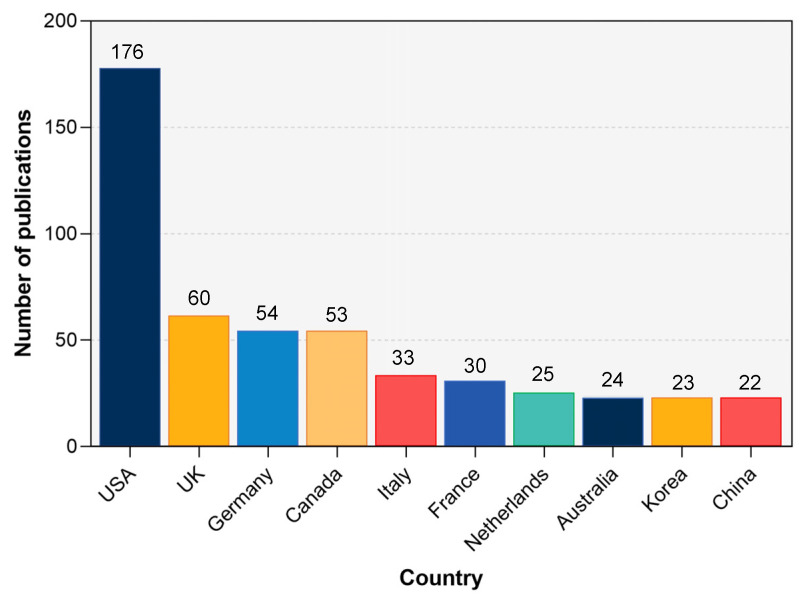
The top 10 most contributive countries within rhythmic auditory stimulation research. Alt text: A column graph demonstrates the ten most productive countries within rhythmic auditory stimulation research. The United States accounted for around 30% of total publications.

**Figure 3 ijerph-20-00215-f003:**
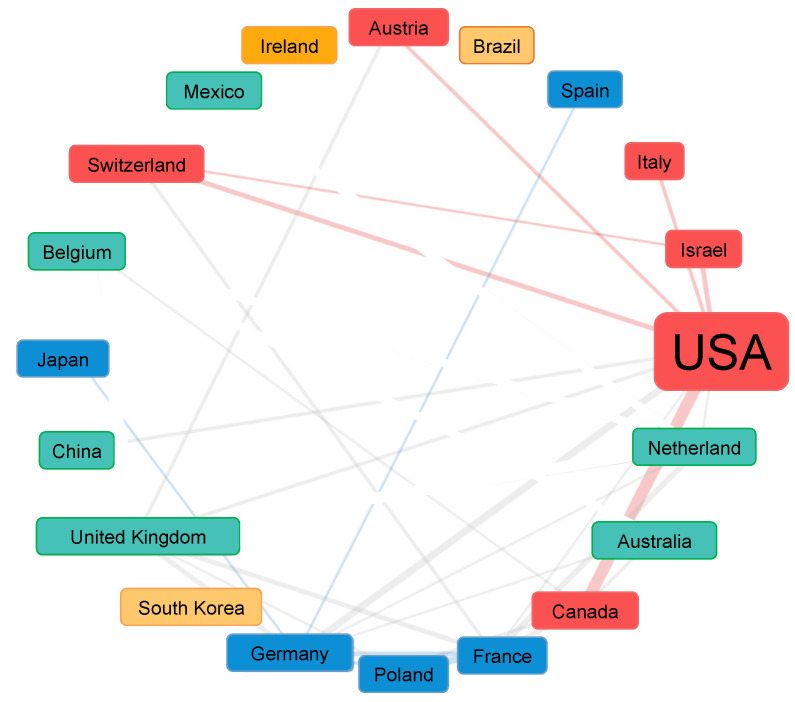
Country collaborations in rhythmic auditory stimulation research. Alt text: A network plot demonstrates the international collaborations within rhythmic auditory stimulation research. The United States has collaborated closely with most countries.

**Figure 4 ijerph-20-00215-f004:**
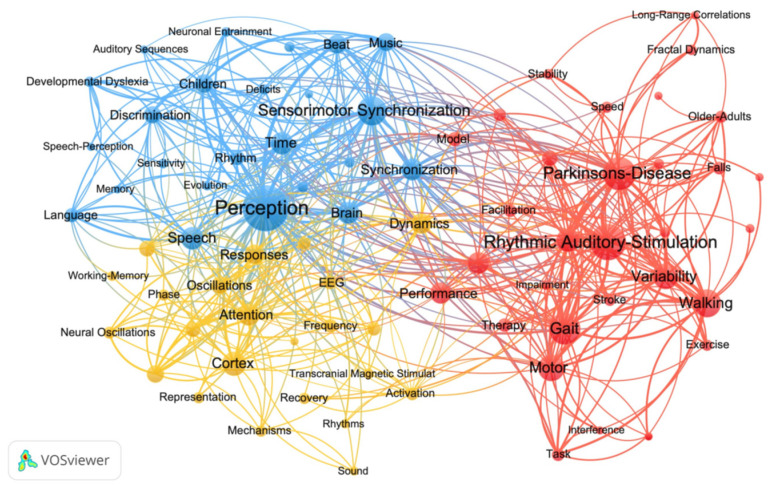
Visualization of the keywords co-occurrence analysis on rhythmic auditory stimulation research. Alt text: A clustering graph is plotting the research topics within rhythmic auditory stimulation research. There are three themes: motor dysfunction, cognition, and perception.

**Figure 5 ijerph-20-00215-f005:**
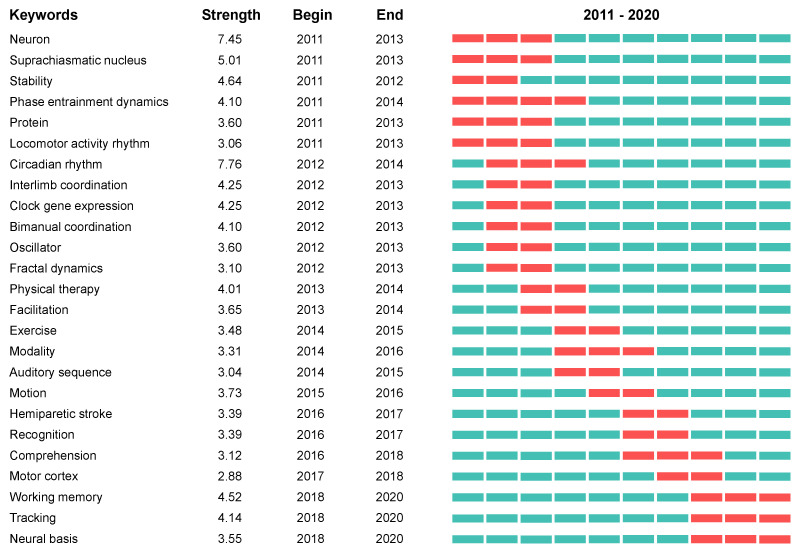
Keywords with the strongest citation bursts of publications on rhythmic auditory stimulation research. Alt text: The top 25 keywords with the strongest citation bursts of publications on rhythmic auditory stimulation research among 2011–2020. Working memory, tracking, and neural basis were the most recent three.

**Figure 6 ijerph-20-00215-f006:**
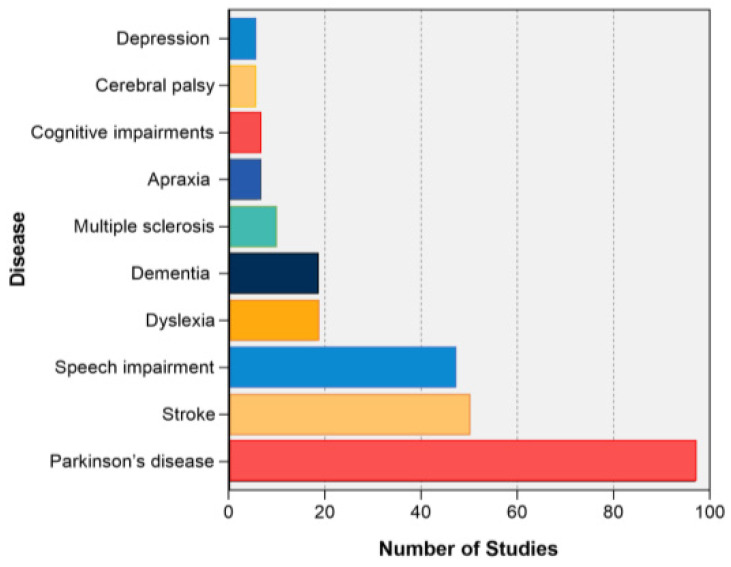
Top 10 most-focused diseases in rhythmic auditory stimulation research. Alt text: A bar chart demonstrates the ten most focused diseases within rhythmic auditory stimulation research between 2001–2020. Parkinson’s disease, stroke, and speech impairment received most attention for the 20 years.

**Table 1 ijerph-20-00215-t001:** The top 10 journals that published articles on rhythmic auditory stimulation research.

Sources	Np	IF 2020	Category	Region
PLoS ONE	32	3.240	Multidisciplinary	USA
Frontiers in Neuroscience	20	4.677	Neuroscience	Switzerland
Experimental Brain Research	16	1.972	Neuroscience	Germany
Frontiers in Human Neuroscience	16	3.169	Neuroscience	Switzerland
Human Movement Science	16	2.161	Psychology	The Netherlands
Journal of Neuroscience	16	6.167	Neuroscience	UK
Neuroimage	16	6.556	Neuroimaging	USA
Frontiers in Neurology	15	4.003	Neuroscience	Switzerland
European Journal of Neuroscience	14	3.386	Neuroscience	UK
Gait & Posture	14	2.840	Rehabilitation	Ireland

Note. Np = number of publications.

**Table 2 ijerph-20-00215-t002:** The top 10 institutions of origin of rhythmic auditory stimulation research.

	Institution	Country	Np	Publications, %
1	University of Toronto	Canada	42	7.17%
2	Colorado State University	USA	26	4.44%
2	Radboud University Nijmegen	The Netherlands	26	4.44%
4	University of Cambridge	UK	25	4.27%
5	Northwestern University	USA	22	3.75%
5	Washington University	USA	22	3.75%
7	McMaster University	Canada	21	3.58%
7	University of Maryland	USA	21	3.58%
9	University of California Irvine	USA	20	3.41%
10	Max Planck Institute for Human Cognitive and Brain Sciences	Germany	18	3.07%

Note. Np = Number of publications.

**Table 3 ijerph-20-00215-t003:** Top 10 of most productive authors publishing on rhythmic auditory stimulation.

	Author	Country	Np
1.	Goswami U.	UK	15
2.	Thaut M.	Canada	13
3.	Dalla-Bella S.	Canada	12
4.	Nieuwboer A.	Belgium	10
5.	Rochester L.	UK	10
6.	Kotz S.	Germany	9
7	Trainor L.	Canada	9
8.	Grahn J.	Canada	8
9.	Kwakkel G.	The Netherlands	8
10.	Earhart G.	USA	7

Note. Np = Number of publications.

**Table 4 ijerph-20-00215-t004:** The top 10 most-cited articles during 2001–2020.

	Paper	Autor(s)	Year	Journal	DOI
1.	Rhythm and beat perception in motor areas of the brain	Grahn & Brett	2007 [21]	Journal of Cognitive Neuroscience	10.1162/jocn.2007.19.5.893
2.	Repetitive bilateral arm training and motor cortex activation in chronic stroke: A randomized controlled trial	Luft et al.	2004 [22]	Journal of the American Medical Association	10.1001/jama.292.15.1853
3.	Auditory closed-loop stimulation of the sleep slow oscillation enhances memory	Ngo et al.	2013 [23]	Neuron	10.1016/j.neuron.2013.03.006
4.	Internalized timing of isochronous sounds is represented in neuromagnetic beta oscillations	Fujioka et al.	2012 [24]	Journal of Neuroscience	10.1523/jneurosci.4107-11.2012
5.	Wearable assistant for Parkinson’s disease patients with the freezing of gait symptom	Bachlin et al.	2010 [25]	IEEE Transactions on Information Technology in Biomedicine	10.1109/titb.2009.2036165
6.	Rhythmic auditory stimulation modulates gait variability in Parkinson’s disease	Hausdorff et al.	2007 [26]	European Journal of Neuroscience	10.1111/j.1460-9568.2007.05810.x
7	Rhythmic auditory stimulation improves gait more than NDT/Bobath training in near-ambulatory patients early poststroke: A single-blind randomized trial	Thaut et al.	2007 [27]	Neurorehabilitation and Neural Repair	10.1177/1545968307300523
8.	To musicians, the message is in the meter: Pre-attentive neuronal responses to incongruent rhythm are left-lateralized in musicians	Vuust et al.	2005 [28]	Neuroimage	10.1016/j.neuroimage.2004.08.039
9.	Cortically projecting basal forebrain parvalbumin neurons regulate cortical gamma band oscillations	Kim et al.	2015 [29]	Proceedings of the National Academy of Sciences	10.1073/pnas.1413625112
10.	Rhythmic motor entrainment in children with speech and language impairments: Tapping to the beat	Corriveau & Goswami	2009 [30]	Cortex	10.1016/j.cortex.2007.09.008

**Table 5 ijerph-20-00215-t005:** The top 10 most-cited articles during 2016–2020.

	Paper	Author(s)	Year	Journal	DOI
1.	Motor origin of temporal predictions in auditory attention	Morillon & Baillet	2017 [31]	Proceedings of the National Academy of Sciences	10.1073/pnas.1705373114
2.	Effect of rhythmic auditory cueing on parkinsonian gait: A systematic review and meta-analysis	Ghai et al.	2018 [32]	Scientific Reports	10.1038/s41598-017-16232-5
3.	Distinct β band oscillatory networks subserving motor and cognitive control during gait adaptation	Wagner et al.	2016 [33]	Journal of Neuroscience	10.1523/JNEUROSCI.3543-15.2016
4.	Selective entrainment of theta oscillations in the dorsal stream causally enhances auditory working memory performance	Albouy et al.	2017 [34]	Neuron	10.1016/j.neuron.2017.03.015
5.	Gait improvement via rhythmic stimulation in Parkinson’s disease is linked to rhythmic skills	Dalla-Bella et al.	2017 [35]	Scientific Reports	10.1038/srep42005
6.	EEG oscillations entrain their phase to high-level features of speech sound	Zoefel & Vanrullen	2016 [36]	Neuroimage	10.1016/j.neuroimage.2015.08.054
7	Music and dyslexia: A new musical training method to improve reading and related disorders	Habib et al.	2016 [37]	Frontiers in Psychology	10.3389/fpsyg.2016.00026
8.	Temporal prediction in lieu of periodic stimulation	Morillon et al.	2016 [38]	Journal of Neuroscience	10.1523/JNEUROSCI.0836-15.2016
9.	Movement sonification: Effects on motor learning beyond rhythmic adjustments	Effenberg et al.	2016 [39]	Frontiers in Neuroscience	10.3389/fnins.2016.00219
10.	An oscillator model better predicts cortical entrainment to music	Doelling et al.	2019 [40]	Proceedings of the National Academy of Sciences	10.1073/pnas.1816414116
	Music supported therapy promotes motor plasticity in individuals with chronic stroke	Ripollés et al.	2016 [41]	Brain Imaging and Behavior	10.1007/s11682-015-9498-x

## Data Availability

Data are available upon reasonable request from the authors.

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
