# Peer review of "Mapping Research Trends from 20 Years of Publications in Rhythmic Auditory Stimulation"

_ijerph, 2022, doi:10.3390/ijerph20010215_

Round 1

Reviewer 1 Report

This bibliometric analysis study chose to provide a survey analysis of the literature on rhythmic auditory stimulation. Overall, the article presents nice tables and graphs from the use of appropriate software packages for bibliometric analysis and straight interpretations of the literature based on the tables and graphs. The introduction and discussion sections could be further strengthened to provide the motivation and integrated assessment.

1. In the introduction, please highlight existing review papers (including meta-analysis on the effects of rhythmic auditory stimulation and their findings. Some links are provided here:

Effect of rhythmic auditory cueing on parkinsonian gait: A systematic review and meta-analysis | Scientific Reports (nature.com)

Effectiveness of Rhythmic Auditory Stimulation on Gait in Pa... : Holistic Nursing Practice (lww.com)

Then point out what a bibliometric analysis can add to the existing reviews. Please emphasize the motivations for doing the bibliometric analysis. 

2. In the discussion, the sections on motor dysfunction, cognition and perception need to point out the clinical populations that have used RAS in various investigations. Some limitations and future directions need to be provided after brief coverage of these areas. 

3. The conclusions part needs to have its own title. It appears abrupt to have conclusions right after 4.1-4.3 without some integrated discussion with respect to the existing findings and unresolved issues. Are the perceptual, cognitive and motor skills independent of each other in these studies? Exactly why arethe The articles, Grahn & Brett [43] and Morillon & Baillet [44], worthy of attention? More expanded discussion can be helpful for the readers. 

Author Response

Reviewer 1:

This bibliometric analysis study chose to provide a survey analysis of the literature on rhythmic auditory stimulation. Overall, the article presents nice tables and graphs from the use of appropriate software packages for bibliometric analysis and straight interpretations of the literature based on the tables and graphs. The introduction and discussion sections could be further strengthened to provide the motivation and integrated assessment.

  1. In the introduction, please highlight existing review papers (including meta-analysis on the effects of rhythmic auditory stimulation and their findings. Some links are provided here:

Effect of rhythmic auditory cueing on parkinsonian gait: A systematic review and meta-analysis | Scientific Reports (nature.com)

Effectiveness of Rhythmic Auditory Stimulation on Gait in Pa... : Holistic Nursing Practice (lww.com)

Then point out what a bibliometric analysis can add to the existing reviews. Please emphasize the motivations for doing the bibliometric analysis. 

Response:

       Thanks for the valuable suggestions. The introduction has been updated. Please check the line 27-46.

  1. In the discussion, the sections on motor dysfunction, cognition and perception need to point out the clinical populations that have used RAS in various investigations. Some limitations and future directions need to be provided after brief coverage of these areas. 

Response:

       Thanks for the great suggestions. The discussion, direction and limitation have been updated (line 231-261).

  1. The conclusions part needs to have its own title. It appears abrupt to have conclusions right after 4.1-4.3 without some integrated discussion with respect to the existing findings and unresolved issues. Are the perceptual, cognitive and motor skills independent of each other in these studies? Exactly why arethe The articles, Grahn & Brett [43] and Morillon & Baillet [44], worthy of attention? More expanded discussion can be helpful for the readers. 

Response:

       The conclusion has been updated based on the reviewer’s suggestions.

Reviewer 2 Report

I thank tha authors for submitting their manuscript. Here are some comments:

- The abstract should not be structured.

- Avoid acronyms that have not been identified or indicate it (remember that the summary is an independent unit).

- The keywords, as far as possible MeSH terms and the are not repeated with those used in the title.

- The introduction requires more work. Recent literature shows benefits of the RAS, will there be previous systematic reviews or studies that synthesize the literature, what practical implication does this review have? Is it important to do it?

- In methodology, some basic study to carry out the data collection procedure? What authors were they based on?

- The search is only un WoS, in the summary it cause confusion to mix databases with manager programs!!!

- The terms selected for the search following any recommendation? MeSH terms, thesaurus or free terms?

- The results are very well presented, I only suggest: In Figure 2 add the number to each bar and for Tables 4 and 5 indicate the full name of the journals.

- The conclusion must be separated and indicated with a title, the first paragraph would be left for the end of the conclusion to start with the answer to the aim. On the other hand, eliminate the cites since the conclusion is referring to you review!!!

Author Response

I thank tha authors for submitting their manuscript. Here are some comments:

- The abstract should not be structured.

Response:

       Updated (line 12-23), thanks!

- Avoid acronyms that have not been identified or indicate it (remember that the summary is an independent unit).

- The keywords, as far as possible MeSH terms and the are not repeated with those used in the title.

Response:

       Updated (line 24), thanks!

- The introduction requires more work. Recent literature shows benefits of the RAS, will there be previous systematic reviews or studies that synthesize the literature, what practical implication does this review have? Is it important to do it?

Response:

       Thanks for the valuable suggestions. The introduction has been updated. Please check the line 27-46.

- In methodology, some basic study to carry out the data collection procedure? What authors were they based on?

Response:

       No basic work was referred to the present study. Few data analytic tools were applied and have been cited (line 61-75). Thanks for pointing out this question.

- The search is only un WoS, in the summary it cause confusion to mix databases with manager programs!!!

Response:

       Added to line 257-263. Thanks for reminding this!

- The terms selected for the search following any recommendation? MeSH terms, thesaurus or free terms?

Response:

       No mesh term was available to the present study, we adopted free term searching. We’ve clarified in text (line 50). Thanks for reminding this!

- The results are very well presented, I only suggest: In Figure 2 add the number to each bar and for Tables 4 and 5 indicate the full name of the journals.

Response:

       Thanks for the kind comments! The figure and tables have been updated as the reviewer suggested.

- The conclusion must be separated and indicated with a title, the first paragraph would be left for the end of the conclusion to start with the answer to the aim. On the other hand, eliminate the cites since the conclusion is referring to you review!!!

Response:

Updated, thanks for the explanations!

Round 2

Reviewer 1 Report

I have read the responses and revision. I recommend acceptance. 

Reviewer 2 Report

Thank to the autors for new version of the manuscript.